# Ecological Parameters of Water Bodies in the Northern Part of the Upper Volga Region with River Flow Regulations

Bernard Gałka [1], Alexandra Novak [2], Mikhail Novak [3], Dmitry Vinogradov [4] and Ryszard Polechoński [5,*]

1 Institute of Soil Science and Environmental Protection, Wrocław University of Environmental and Life Sciences, ul. Grunwaldzka 53, 50357 Wrocław, Poland; bernard.galka@upwr.edu.pl
2 Department of Microbiology, Ryazan State Medical University Named after Academician I.P. Pavlov, Ministry of Health of the Russian Federation, Vysokovolnaya Str., 9, 390026 Ryazan, Russia; marieta69@mail.ru
3 Department of Epidemiology, Ryazan State Medical University Named after Academician I.P. Pavlov, Ministry of Health of the Russian Federation, Vysokovolnaya Str., 9, 390026 Ryazan, Russia; peace100@mail.ru
4 Department of Agronomy and Agrotechnologies, Ryazan State Agrotechnological University Named after P.A. Kostychev, Kostychev Str., 1, 390044 Ryazan, Russia; vdvrzn@mail.ru
5 Institute of Animal Breeding, Wrocław University of Environmental and Life Sciences, ul. Chełmońskiego 38C, 51630 Wrocław, Poland
* Correspondence: ryszard.polechonski@upwr.edu.pl

**Abstract:** The regulation of river flow in the Volga basin has caused irreversible changes to aquatic ecosystems. The transformation of the Volga into a cascade of hydraulic structures with a non-flow regime has resulted in a decrease in depth and flow, and an increase in the temperature and concentration of chemical elements, which has induced the process of eutrophication. The change in the species diversity of aquatic organisms under conditions of intense eutrophication was studied on models of water bodies from the Volga basin; the Kostroma section of the Gorky reservoir (Kostroma spill and the middle river section), and lakes Galichskoe and Chukhlomskoe were studied. Rheophilic biocenosis was replaced by a limnophilic one, the migration paths of fish were disrupted, and population characteristics were changed. In accordance with environmental conditions, the level of primary production and the calculated Carlson trophic index (TSI) and Broth-proposed index (ITS) (1987), the water bodies of the northern part of the upper Volga region are classified as follows: the middle river section of the Gorky reservoir is mesotrophic-eutrophic (TSI = 55.2, ITS = 16.2); the Kostroma spill is eutrophic with a tendency to hypertrophy (TSI = 67.4, ITS = 6.8); Lake Galichskoe is eutrophic with a tendency to dystrophy (TSI = 63.2, ITS = 8.4), and Lake Chukhlomskoe is hypertrophic with a tendency to dystrophy (TSI = 77.4, ITS = 8.0). In addition, frequent fluctuations in water level, reaching 1 m, have had an adverse effect on inhabitants of the littoral zone including the spawning fish, which may lead to disappearance of some of the region's most sensitive species.

**Keywords:** Volga basin; hydro construction; environmental parameters; eutrophication; hydrobionts

## 1. Introduction

Due to the regulation of large river systems, the discharge of insufficiently treated domestic, industrial and agricultural wastewater leads to the significant accumulation of nutrients in reservoirs. Their trophic status changes, water quality decreases and, ultimately, populations and species diversity in phyto- and zoocenoses decrease.

Large-scale hydro construction in the second half of the 20th century has led to the radical transformation of river systems in the Russian Federation. Changes in flow velocity, depth, oxygen, temperature, chemical regimes, and other abiotic factors have contributed to the accumulation of bottom sediments, bed silting, increased eutrophication, disruption of the structure of individual hydrobiont populations, and some changes in the existing dynamic equilibrium of biocenoses [1].

The largest river in the European part of the Russian Federation is the Volga. The environmental conditions of the Volga have changed significantly since a cascade of nine

reservoirs were created (Rybinsk, Gorky, Cheboksary and others). In fact, parts of the Volga river have ceased to exist, and become a low-flow body of water of the lake-river type.

Significant territories were spilled when constructing dams, which led to the leaching of nutrients from the soil, the decomposition of vegetation in water-spilled areas, eutrophication and a change in the species composition of biocenoses.

Eutrophication is caused by excessive nutrients in the water reservoir, mainly nitrogen and phosphorus compounds. The consequence is the "fertility effect", or the enhanced development of plant organisms. Macrophytes and phytoplankton consume oxygen intensively, as a result of this, unfavorable conditions for fish are formed, and they are sometimes killed. The decomposition of the bodies of dead animals is accompanied by additional oxygen consumption and the release of methane and carbon dioxide.

Since the 1970s, the problem of eutrophication has acquired a global scale awareness and fundamental importance, due to its negative consequences which can lead to a complete loss of the biosphere functions of aquatic ecosystems. The increasing intensity of eutrophication processes and their specificity in various types of water require the development of special methods to monitor and protect aquatic ecosystems [2]. Reservoirs as ecological systems transformed by man are a model for studying the mechanisms of homeostasis of biocenoses in conditions of changing natural complexes. Violating the flow of reservoirs leads to a radical restructuring of the ichthyocenosis: the rheophilic complex is replaced by a limnophilic one [3].

Additional impacts of reservoir construction include the creation of a new pelagic habitat, the replacement of spill plains with lacustrine littoral and sublittoral habitats, the creation of a complex bathyal habitat from former river channels, and the replacement of river flow patterns by pelagic water mass circulations. Populations of rheophilic species have declined, whereas a new pelagophilic fish guild has developed [4]. Commercial fish catches decrease after damming, both in the main channel and in spill plain lakes. All catches are dominated by species with a eurytopic flow preference, although catches from the main channel contain more rheophilic species, and spill plain catches contained more limnophilic and phytophilic species [5]. It was found that habitat quality was significantly poorer in artificial pools created above dams than all other sampling sites. Fast riffle specialist taxa were most abundant in high-quality riffle habitats, being farthest from the dams, whereas fast generalists and pelagophils were largely restricted to areas below the downstream. These dams played a substantial role in shaping habitat, which impacted the fish community on a functional level. Utilizing this functional approach has enabled researchers to link the effects of impoundments to the structure of fish communities, and form generalizations that can be applied to other systems [6]. Similar changes in abiotic parameters and composition of hydrobiocenoses after the construction of dams on flat rivers were observed by researchers from Europe, Asia, and the USA [7–9].

The infection of fish with parasites has increased due to unfavorable changes in the hydrochemical and gas regimes of aquatic ecosystems under the influence of civilization, and due to an increase in populations of intermediate hosts of helminths (annelids, crustaceans, mollusks). Insufficiency and inferiority of feed have contributed to reducing the resistance of fish populations. The consequence of a multifactorial effect, including the influence of helminths, parasitic protozoa and crustaceans, is the loss of biological productivity in the reservoir, due to slowing down the development of and the consequent death of fish. It is important to determine the causes and consequences of an increase in the concentration of organic and mineral substances in reservoirs. Significant changes in ecological parameters of reservoirs (depth, flow, temperature, and concentration of chemical elements) induce succession processes in the established biocenoses. The construction of a dam on Lake Chukhlomskoe led to the disruption of its connection with the upper Volga basin, and the intensity of fish migration significantly decreased. The factor of relative isolation of a local herd of fish in a limited water area contributed to the emergence of a set of characteristics in individuals that reliably distinguish this local population from neighboring ones [10]. Since lakes Galichskoe and Chukhlomskoe have the same type of sapropel

reservoirs (shallow, eutrophic, close in area), Polovkova and Nadirov [10] aimed to identify how strongly the degree of isolation and violation of migrations affected the biological features (morphology, growth, nutrition, etc.) of the fish inhabiting these reservoirs. It was found that the connection between lake and river fish populations in the studied reservoirs was broken. The composition of the ichthyocenosis and main biological indicators of fish in Lake Chukhlomskoe, completely isolated from the river system, underwent the most pronounced changes. The purpose of the research was to study the ecological parameters of the Volga and the lakes of the Volga basin (Galichskoe and Chukhlomskoe), 50 years after the creation of the reservoir. For the first time in the late 20th–early 21st century, comprehensive studies characterizing the hydrobiocenoses of the Volga basin in conditions of long-term regulation of the river flow were carried out. For the first time, the size and age indicators of the population of the dominant species in the upper Volga basin, *Abramis brama*, were studied. A complex of factors causing an increase in the level of primary production in the reservoirs of the northern part of the upper Volga region were studied. The intensification of eutrophication processes is currently characteristic of many reservoirs of the European part of the Russian Federation. The leading factor in increasing the level of primary production is mainly anthropogenic impact, in particular, the overregulation of large river systems. The transformation of the biocenosis under conditions of increased concentration of biogenic elements occurs in a similar way in marine and freshwater ecosystems, and allows the identification of indicator species, the study of which has both theoretical and applied significance.

## 2. Materials and Methods

The research scheme was as follows:

| Research object | |
|---|---|
| Kostroma section of the Gorky reservoir (Kostroma spill and the middle river section) | lakes Galichskoe and Chukhlomskoe |

| The volume of studies in 1999-2005 was as follows |
|---|
| 7358 fish specimens were studied, including *Abramis brama L.*—5683, *Blicca bjoerkna L.*—478, *Rutilus rutilus L.*—118, *Abramis ballerus L.*—519, *Pelecus cultratus L.*—29, *Lucioperca (Sander) lucioperca L.*—42, *Esox lucius L.*—304, *Aspius aspius L.*—22, *Perca fluviatilis L.*—31, *Leuciscus idus L.*—39, *Carassius carassius L.*—45, *Tinca tinca L.*—36, *Leucaspius delineatus Haeckel*—12 |

| Number of studies conducted |
|---|
| 7358 fish surveys; ageing of 5650 fish specimen, of which 1470 by rays; 32 by bottom trawls; 178 by pelagic trawls; 130 by trap-net fishing in the Kostroma spill, 121 in lake Galichskoe; 15 in lake Chukhlomskoe and 68 by drift net fishing in lake Galichskoe |

| Ecological characteristics of reservoirs: |
|---|
| abiotic parameters; productivity; population characteristics of the ichthyofauna; trophic status |

The change in the species diversity of aquatic organisms under conditions of intense eutrophication was studied on models of water bodies of the Volga basin, with specific reference to the Kostroma section of the Gorky reservoir (Kostroma spill and the middle river section), and lakes Galichskoe and Chukhlomskoe.

The type of fish was determined, their length was measured, and weighing was carried out using electronic scales. The age of the fish up to three years was determined by the number of circularly arranged segments on the scales, taken 3–5 from each specimen above the lateral line of the body, at the level of the second ray of the dorsal fin. The age of fish older than three years was determined by the number of rings on the slice of the first ray of the dorsal fin. Before microscopic examination using a binocular stereoscopic microscope, the scales were moistened with water, and the ray sections (0.5–1 mm thick) were moistened with 10% ammonia solution. "The Gorky reservoir is located on the Volga River, between the cities of Rybinsk and Gorodets. It was created in 1957. Its length is 434 km, and its width is up to 14 km. The maximum depth is 22 m, although the average depth is 6.4 m, and its volume is 8.8–10.3 km$^3$. The reservoir area is 157 thousand ha. The largest tributaries are the Kostroma and Unzha rivers. The shallow zone with the depth of up to 2 m constitutes 18% of the area (28 thousand ha). About 13% of the area of shallow water is used by fish for spawning. The reservoir is divided into five main sections: the upper river; the Kostroma spill; the middle river section; the lake-shaped and the dam areas. The studies were carried out within the Kostroma region, and the environmental characteristics of the Kostroma spill and the middle river section (the channel part of the Gorky reservoir) were studied. The Kostroma spill is a lake-type reservoir, formed as a result of the spilling of Kostroma lowland after the construction of the dam and the creation of a reservoir. Its area is 26 thousand hectares, the maximum depth is 8.0 m and the average depth is 3–4 m. The bottom is silty and peat. The flowage is low. Reproduction and feeding periods in the Kostroma section of the Gorky reservoir have taken place after the spill. The middle river section is the section of the Gorky reservoir from the city of Kostroma to the Elnat River. The area is 30.2 thousand ha, the maximum depth is 17 m and the average one is 7–10 m. The flow velocity is 0.15 m/s. The cities of Kostroma and Volgorechensk are located in this part of the reservoir" [11]. The shallow zone is 3.3 thousand ha. The bottom in the channel part is represented by dense sand, and that on the flowage land is meadow soil, with some deposit of spring spills. "The mouth reach of the tributaries Keshka, Sunzha, Mesa, Koldoma and the expansion with multiple islands (located between villages Gustomesovo and Krasnoe-on-the Volga) are favorable places for fish reproduction and feeding. Lake Galichskoe is the main fishing reservoir in the region with a diverse fish species composition and bream dominance. The lake has an oval shape, its length is 16.7 km, the maximum width is 6.4 km, the maximum depth is 5.0 m, the average depth is 1.75 m and the lake area is 75.4 km$^2$. Several rivers flow into lake Galichskoe and the largest ones are the Chelsma and the Srednyaya. The Veksa (the left tributary of the Kostroma River) flows out. The area of Lake Chukhlomskoe is 48.7 km$^2$. The lake has a close to round shape, 6–7 km across. The depth is up to 4.5 m. The shores of the reservoir are flat and swampy; the bottom is muddy. The lake is fed by surface runoff and groundwater. There are 17 streams and rivers flowing into it, the Veksa River (a tributary of the Vocha River) flows out" [11].

The trophic status of reservoirs was determined using a set of parameters: in accordance with the environmental classification of Tineman and Naumann, the level of trophy, abiotic factors (depth, transparency of the reservoir, pH, the presence of hypolimnial oxygen, biogen, etc.), the geographical location of the reservoir and the nature of the catchment were considered. The value of primary production was indicated according to Nizhny Novgorod laboratory GOSNIORKH.

Carlson's trophic index was calculated by the following formula:

$$TSI = 10\ (6\ log_2\ SD) \tag{1}$$

where *SD* is transparency [12].

On the basis of considering the integral indicators of R.E. Carlson (1977), a numerical classification scale was compiled (Table 1). The calculations of the Carlson trophic index (*TSI*) are based on close correlations between the parameters of the aquatic environment—transparency, chlorophyll concentration (Chl "a" *, mg/m$^3$) in water and the total phosphorus content (P$_{total}$ *, mg/m$^3$).

**Table 1.** Trophic index and related parameters (by Carlson, [12]).

| Type of Reservoir | *TSI* | Transparency, m | P$_{total}$ *, mg/m$^3$ | Chl "a" *, mg/m$^3$ |
|---|---|---|---|---|
| Oligotrophic | 0 | 64 | 0.75 | 0.04 |
| | 10 | 32 | 1.5 | 0.12 |
| | 20 | 16 | 3 | 0.34 |
| | 30 | 8 | 6 | 0.94 |
| Mesotrophic | 40 | 4 | 12 | 2.6 |
| | 50 | 2 | 24 | 6.4 |
| Eutrophic | 60 | 1 | 48 | 20 |
| | 70 | 0.5 | 96 | 56 |
| | 80 | 0.25 | 192 | 154 |
| Hypereutrophic | 90 | 0.12 | 384 | 427 |
| | 100 | 0.062 | 768 | 1183 |

* In the surface layer of water.

Broth [13] proposed an index (*ITS*) calculated from the concentration of chlorophyll (*C*) in water:

$$ITS = 40 - 20 \, lgC \qquad (2)$$

## 3. Results and Discussion

The Gorky reservoir belongs to the type of lake-river reservoirs. The fish fauna have developed from species dwelling both in the Volga riverbed and the rivers flowing into it. The fish fauna were dominated by typical limnophylics: bream (*Abramis brama* L.), white bream (*Blicca bjoerkna* L.), roach (*Rutilus rutilus* L.), perch (*Perca fluviatilis* L.) and pike (*Esox lucius* L.). Representatives of the *Cyprinidae* family were also found: blue bream (*Abramis ballerus* L.), bleak (*Alburnus alburnus* L.), crucian carp (*Carassius carassius* L.), verkhovka (*Leucaspius delineatus* Haeckel), ide (*Leuciscus idus* L.) and those of the *Percidae* family: pike perch (*Lucioperca (Sander) lucioperca* L.), ruff (*Acerina cernua* L.). Typical rheophils: eelpout (*Lota lota* L.) and asp (*Aspius aspius* L.) had a low abundance. Against the background of the regulation of the Volga when creating the Gorky reservoir in the upper sections of the river, the number of razor fish (*Pelecus cultratus* L.) decreased.

According to the results of industrial fishing in the Kostroma section of the Gorky reservoir, the age structure of the population of the dominant fish species *Abramis brama* was studied (Table 2). "The most numerous are the generations of 2, 3, 4 and 5 years. The number of 6–11 years old fish is 48.2% less than that of two-year or five-year-old fish. A decrease in the fish population, starting from the age of 6, confirms a rather high level of natural mortality at the sixth to seventh years of life" [11].

A similar age structure is characteristic of fish populations with an average life cycle. The close parameters of the number of 6–8 years old generations in the *Abramis brama* populations in the middle river section, and the Kostroma spill, confirm their fairly stable state in the Kostroma section of the Gorky reservoir.

The quantitative ratio of different fish species in the water bodies of the northern part of the upper Volga region make it possible to judge the results of trawling on the Kostroma section of the Gorky reservoir: large bream (more than 30 cm)—3.4 tons; medium bream (21–30 cm)—13.9 tons; small bream (15–20 cm)—14 t (1:4:4); roach—7.5 ton; blue bream—2.5 tons; razor fish—0.3 tons; ide—0.07 tons; pike—2 tons; pike perch—0.8 tons (Figure 1).

**Table 2.** Size and age structure of the *Abramis brama* bream population in the Kostroma section of the Gorky reservoir.

| Age Years | Length (*l*) cm | | | | | | | | | | | | | | | | | | | | | | | | | | | | $l_m$ | n Specimen |
|---|---|---|---|---|---|---|---|---|---|---|---|---|---|---|---|---|---|---|---|---|---|---|---|---|---|---|---|---|---|---|
| | 16 | 17 | 18 | 19 | 20 | 21 | 22 | 23 | 24 | 25 | 26 | 27 | 28 | 29 | 30 | 31 | 32 | 33 | 34 | 35 | 36 | 37 | 38 | 39 | 40 | 42 | 43 | 46 | | |
| 1+ | 8 | 12 | 11 | 7 | 5 | 3 | 1 | 1 | | | | | | | | | | | | | | | | | | | | | 18.4 | 48 |
| 2+ | 2 | 12 | 24 | 27 | 28 | 27 | 12 | 8 | 7 | 2 | 1 | | | | | | | | | | | | | | | | | | 20.0 | 150 |
| 3+ | 2 | 7 | 12 | 20 | 18 | 31 | 12 | 7 | 7 | 4 | 5 | | 2 | 1 | | | | | | | | | | | | | | | 20.8 | 128 |
| 4+ | | 3 | 8 | 11 | 17 | 27 | 11 | 22 | 12 | 10 | 5 | 5 | 7 | 3 | 1 | 2 | 3 | 1 | 1 | | | | | | | | | | 22.8 | 149 |
| 5+ | | | 1 | 1 | 7 | 10 | 12 | 6 | 10 | 8 | 14 | 5 | 6 | 4 | 8 | 5 | 4 | 5 | 6 | 1 | 1 | | | | | | | | 26.1 | 114 |
| 6+ | | | 2 | | 1 | | | 2 | 4 | 5 | 3 | 12 | 5 | 4 | 7 | 5 | 6 | 4 | 6 | 6 | 2 | 2 | 1 | | | | | | 28.8 | 77 |
| 7+ | | | | | | | | 1 | 2 | 2 | 1 | 2 | 3 | 5 | 4 | 4 | 9 | 5 | 11 | 6 | 6 | | | | | | | | 31.6 | 61 |
| 8+ | | | | | | | | | | | | 3 | 1 | 4 | 2 | 3 | 5 | 7 | 10 | 7 | 3 | 3 | | 1 | | 1 | | | 33.1 | 50 |
| 9+ | | | | | | | | | | 1 | 1 | | | 3 | 5 | 2 | 8 | 4 | 2 | 3 | 2 | 1 | 2 | 1 | | | | | 33.4 | 35 |
| 10+ | | | | | | 1 | | | | | 1 | | | 1 | | 2 | 6 | 2 | 6 | 4 | 3 | 1 | | 2 | 1 | | | | 33.5 | 30 |
| 11+ | | | | | | | | | | | | | | | | | 4 | 5 | 8 | 1 | 2 | 2 | | | | | 1 | | 34.3 | 23 |
| 12+ | | | | | | | | | | | | | | | | | | | 6 | 3 | 2 | 1 | | | | | | | 34.8 | 12 |
| 13+ | | | | | | | | | | | | | | | | | | 1 | | 2 | 2 | | | 1 | | | 1 | | 36.7 | 7 |
| 14+ | | | | | | | | | | | | | | 1 | | | | | | | 1 | 2 | | | | | | | 35.8 | 4 |
| 15+ | | | | | | | | | | | | | | | | | | | | 1 | 1 | 1 | 1 | | 1 | | | | 37.2 | 5 |
| 16+ | | | | | | | | | | | | | | | | | | | | | | | | | 1 | 1 | | 1 | 42.7 | 3 |
| 17+ | | | | | | | | | | | | | | | | | | | | | | | | | | | | 1 | 46 | 1 |
| n specimen | 12 | 11 | 34 | 58 | 66 | 76 | 98 | 49 | 47 | 42 | 32 | 31 | 27 | 24 | 22 | 26 | 26 | 39 | 38 | 58 | 33 | 25 | 13 | 5 | 6 | 4 | 2 | 2 | | 897 |

*Source:* (Compiled by the authors).

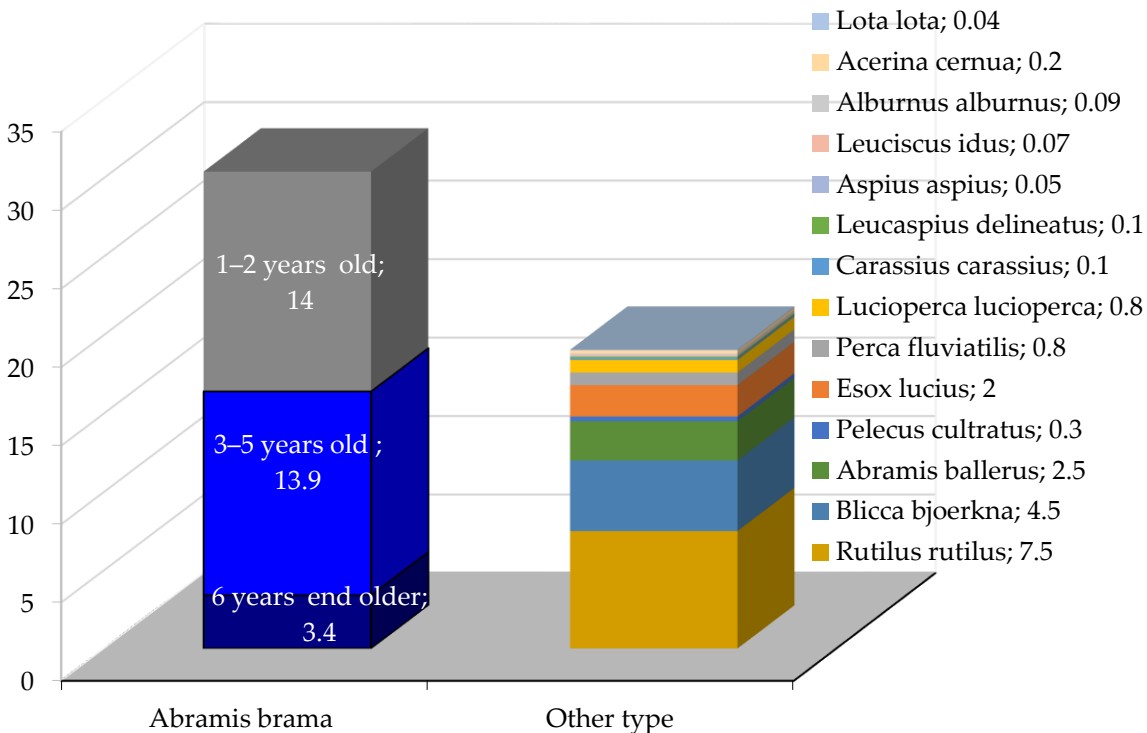

**Figure 1.** The quantitative ratio of various fish species in the Kostroma section of the Gorky reservoir (in tons). *Source*: (Compiled by the authors).

The change in the species structure of the biocenoses was influenced by the creation of the Kostroma State District Power Station in the region of Volgorechensk, which uses the reservoir as a reservoir-cooler. Novak noted that "due to an increase in the temperature of return water by 7 °C and in general in the reservoir area adjacent to the state district power station by 1–1.5 °C, the biomass of phyto- and zooplankton and macrophytes increased significantly causing the increase of food supply for fish" [11]. The latter factor was the reason for the increase in the number of fish subpopulations, plankton feeders and benthophages. For several years, a significant population of fish-eating birds have formed in this area, which are the source of helminth infections for fish.

Novak further noted that "lakes Galichskoe and Chukhlomskoe, which are the largest reservoirs of the upper Volga basin in the Kostroma region, are connected with the Gorky reservoir through the river system. The lakes are fragments of an ancient glacial reservoir that filled the Galichskoe–Chukhlomskoe basin" [11].

Moreover, "the shallowing of the lake during dry climatic periods led to degradation of the catchment network. The territory of Lake Galichskoe and its environment is experiencing slow immersion, river valleys and a lake basin are intensively swamped. Against the background of geological processes and climate change, intensification of biochemical reactions occurs. During wind mixing, biogenic compounds from bottom sediments are involved in the circulation of substances. As a result, the decomposition of organic matter and the further accumulation of sludge are accelerated. The average thickness of the sapropelic silt layer in the basin of Lake Galichskoe is 5.73 m and the volume is 397.6 million $m^3$. Silt lies in a layer up to 15 m on an area of 6.99 thousand ha. The current rate of sapropel accumulation is 25 times higher than the average for the entire history of the lake.

The shores of Lake Galichskoe are flat and swampy. As a result of coastal shallow wave erosion, there are extensions in river mouths that worsen water exchange and intensify macrophyte overgrowth (higher aquatic vegetation occupies 85% of the lake).

Changes in environmental factors during eutrophication of the lake led to a depletion" [11] of the fish species composition (Figure 2).

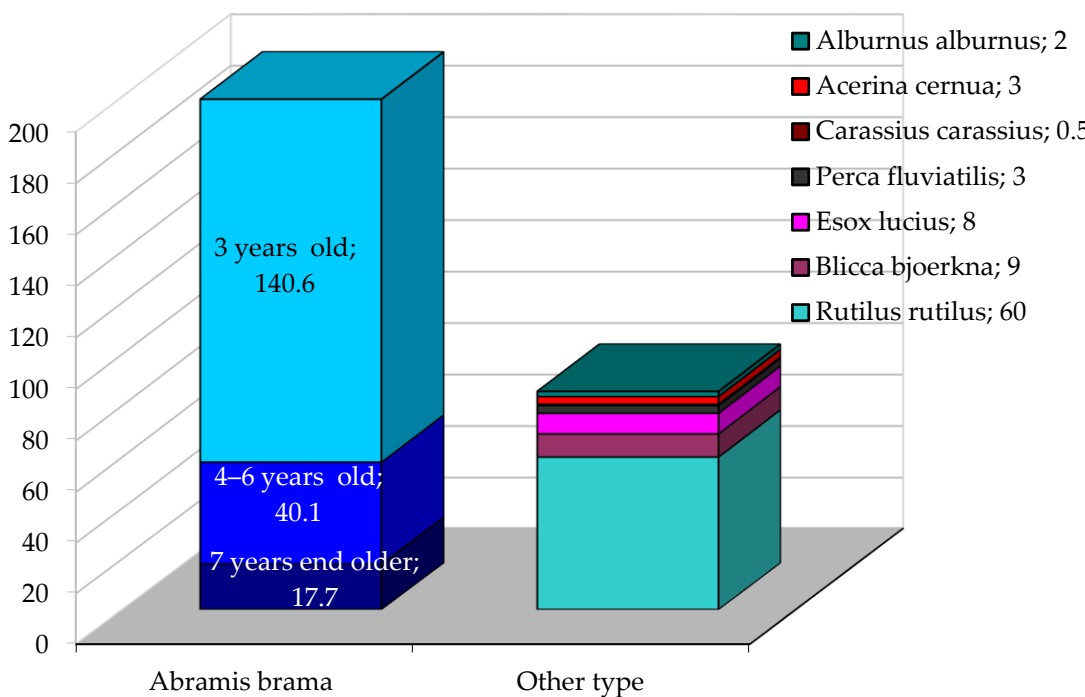

**Figure 2.** The quantitative ratio of various fish species in Lake Galichskoe (in tons). *Source*: (Compiled by the authors).

Bream (*Abramis brama* L.) dominated among fish, roach (*Rutilus rutilus* L.) was abundant and white bream (*Blicca bjoerkna* L.), perch (*Perca fluviatilis* L.), pike (*Esox lucius* L.), crucian carp (*Carassius carassius* L.) were less common. The results of industrial fishing make possible to estimate the age structure of the *Abramis brama* population in Lake Galichskoe. Large bream (over 7 years old) in annual catches equate to 17.7 tons, the medium bream (4–6 years) equate to 40.1 tons, and small bream (3 years) equate to 140.6 tons (1:2:8). Bream trifle (up to 15 cm) are 296 t.

Novak notes that "a significant prevalence of young and small fish in the *Abramis brama* population indicates a high biomass of zooplankton in the food base of Lake Galichskoe and a low abundance of zoo benthos. Juveniles, upon reaching a length of 3 cm, in the presence of a sufficient number of different species of benthos, switch to feeding on small bottom animals. Adult fish is a polyphage, so it feeds on crustaceans, mollusks, worms, insect larvae and algae. But with intensive reproduction of plankton fauna, even adult bream retains predominantly planktonic nutrition. Due to the lower calorific value of zooplankton compared to benthos, the growth rate of *Abramis brama* slows down and some population die due to lack of food" [11].

"Lake Chukhlomskoe, which arose, like Galichskoe, during the second glaciation, is characterized by peculiar environmental parameters. The lake coastline and littoral are overgrown with macrophytes, and the overgrowing area is about 90%. The northeastern part of the reservoir is swampy, and the waterlogging process is progressing. In 1990–1995 low average annual rainfall led to a decrease in surface runoff and the rapid shallowing of the lake began; it receded from the shore by 0.5–1.5 km. In order to restore the initial water level, a dam was built at the exit of the Veksa from the lake. As a result, the depth of lake Chukhlomskoe increased insignificantly, and for a short time. In 1992, under the influence of climatic factors (a dry and hot summer) and also as a result of improper regulation, the water level in the lake sharply decreased, which led to the mass death of fish. A similar situation was repeated in 1998–1999. At present, Lake Chukhlomskoe, with quite pronounced anthropo-pressure (deformation of the coastal zone as a result of the use

of heavy equipment, the creation of an ineffective dam, and the violation of the water-rich character of confluent waterways), is an unbalanced reservoir. The lake has violated the hydrological regime, the ratio of the components of the biocenoses and the processes of natural regulation of the number of hydrobionts. Therefore, even small changes in climatic conditions cause the significant restructuring of biocenoses, with catastrophic consequences for the reservoir" [11]. One of these is a significant decrease in the crucian carp population. As a result, a population of the crucian carp hybrid was formed in the lake due to the fertilization of its eggs with sperm of other species of cyprinids.

Only typical limnophils are recorded in the fish fauna: crucian carp (*Carassius carassius* L.), resistant to the lack of dissolved oxygen, dominate. Roach (*Rutilus rutilus* L.), perch (*Perca fluviatilis* L.), pike (*Esox lucius* L.), verkhovka (*Leucaspius delineatus Haeckel*), tench (*Tinca tinca* L.), ide (*Leuciscus idus* L.), bream (*Abramis brama* L.) and golden carp (*Carassius auratus* L.) are the rarest species (Figure 3).

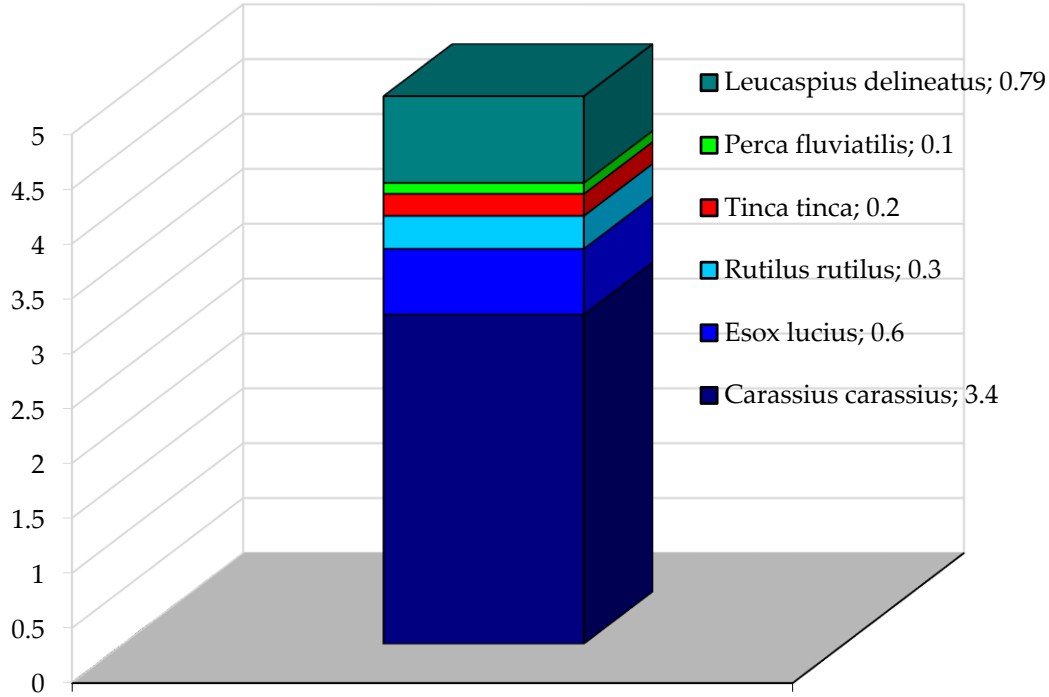

**Figure 3.** The quantitative ratio of various fish species in Lake Chukhlomskoe (in tons). *Source:* (Compiled by the authors).

Despite the insignificant difference in the area of lakes Galichskoe and Chukhlomskoe, the number of populations of various fish species has a significant difference. The species composition of ichthyocenosis and the ratio of different species of fish makes it possible to judge the average annual catch: crucian carp (3.4 tons), pike (0.6 tons), roach (0.3 tons), tench (0.2 tons), perch (0.1 tons) and verkhovka (0.79 tons). Novak identifies that "the relatively low number of fishes is due to the low density of both plankton and benthos. The species composition of fish in various reservoirs in the northern part of the upper Volga region depends on environmental conditions caused by temperature, oxygen and hydrological conditions, food supply, the level of anthropogenic impact and trophic status of the reservoir" [11].

The main environmental characteristics of water bodies in the northern part of the upper Volga region are presented in Figure 4.

The low flow rate in the channel of the Gorky reservoir contributes to the development of phytoplankton that is optimal for maintaining trophic links (production–0.8–1.1 g. wt./m$^2$ per day, 292–401.5 g. wt./m$^2$ per year). This determines the maximum transparency for water bodies in the region (up to 1.5 m) and a high species diversity of aquatic organisms (Figure 4).

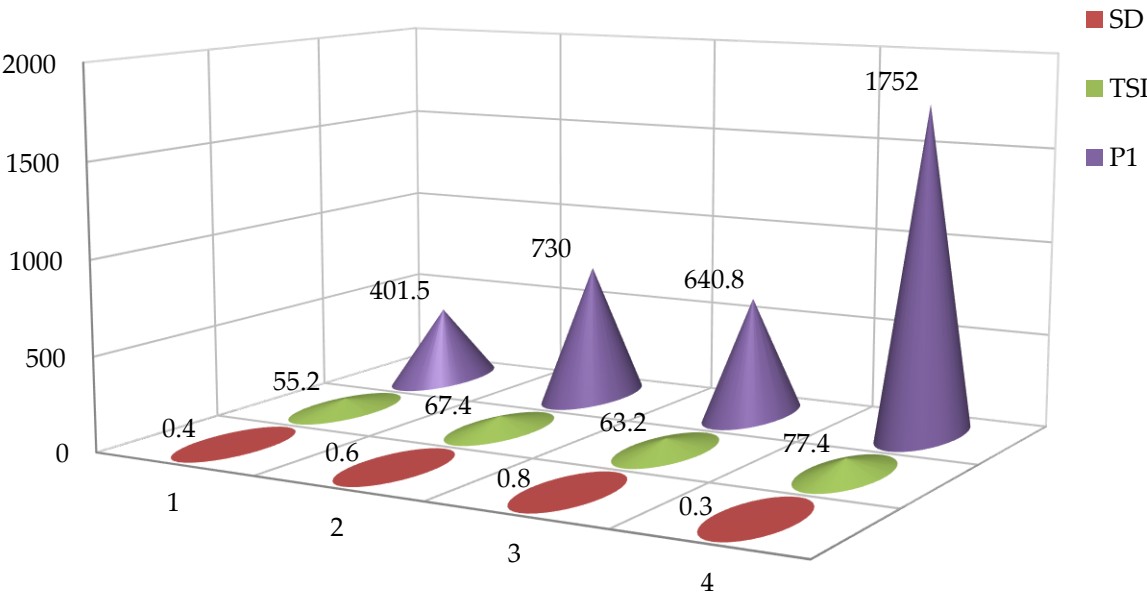

**Figure 4.** Environmental parameters of water bodies in the northern part of the upper Volga region: 1, Gorky reservoir; 2, Kostroma spill; 3, Lake Galichskoe; 4, Lake Chukhlomskoe; SD, transparency, m; P1, level of primary production, g. wt./m$^2$; TSI, Carlson's Trophic Index. *Source*: (Compiled by the authors).

The phytoplankton developed intensively (primary production is 2.0 g. wt./m$^2$ per day, 730 g. wt./m$^2$ per year) in the Kostroma spill, with greater mineralization and an increased content of dissolved nutrients, reducing the transparency of the reservoir to 0.6 m. "As a result, the epilimnion is saturated with oxygen and hypolimnion is characterized by a significant drawback. The reservoir is divided into two clearly demarcated tiers: the upper aerobic and lower anaerobic ones. Favorable conditions for the intensive development of zooplankton and plankton feeders are formed in the upper tier. Organic residues accumulate in the lower tier under anaerobic conditions. As a result, detrital trophic chains become the only ones with oxygen deficiency and an abundance of organics.

Moreover, Lake Galichskoe is characterized by a high content of humic substances and a low level of phytoplankton development compared to the Kostroma spill (primary production 8 g. wt./m$^2$ per day, 640.8 g. wt./m$^2$ per year). Low mineralization, an insignificant amount of dissolved nutrients and an abundant content of aqueous humus are signs of dystrophy. Sapropel, which accumulates in the epilimnion, is formed by hardly soluble humic acids and makes up the bulk of the organic matter in the ecosystem.

A high level of primary production (4.8 g. wt./m$^2$ per day, 1752 g. wt./m$^2$ per year) in Lake Chukhlomskoe is provided by the intensive development of cyanobacteria, with multicellular forms prevailing and forming fairly large trichomes that are excluded from the diet of zooplankton" [11]. This leads to a decrease in the intensity of reproduction of planktonic animals. Due to the lack of zooplankton, the forage base of fish is depleted, and the plankton feeders' biomass decreases. The "dimming of hypolimnion due to the low transparency of the upper layer leads to the death of phytobenthos. As a result, optimal conditions are formed in the lake for detritivorous animals and predators feeding on them. In addition, the abundant development of cyanide causes a high concentration of the metabolites that they release, which are toxic to other organisms. Therefore, the species diversity of aquatic organisms in the lake is extremely low, mainly species with high toxicity survive: tubificid worms, gastropods, crucian carp, tench and pike" [11].

Considering the concentration of chlorophyll in the studied reservoirs, the trophic index for Broth was calculated. The results are presented in Table 3.

**Table 3.** Trophic index of Broth in reservoirs of the Volga basin.

| Area | Chlorophyll Concentration, mg/m$^3$ | *ITS* |
|---|---|---|
| The middle river section of the Gorky reservoir | 15.4 | 16.2 |
| The Kostroma spill | 45.2 | 6.8 |
| Lake Galichskoe | 38.1 | 8.4 |
| Lake Chukhlomskoe | 39.8 | 8.0 |

In accordance with environmental conditions, the level of primary production and the calculated Carlson and Broth trophic indexs (Figure 4), the water bodies of the northern part of the upper Volga region are classified as follows: the middle river section of the Gorky reservoir is mesotrophic-eutrophic (*TSI* = 55.2; *ITS* = 16.2), the Kostroma spill is eutrophic with a tendency to hypertrophy (*TSI* = 67.4; *ITS* = 6.8), Lake Galichskoe is eutrophic with a tendency to dystrophy (*TSI* = 63.2; *ITS* = 8.4), Lake Chukhlomskoe is hypertrophic with a tendency to dystrophy (*TSI* = 77.4; *ITS* = 8.0).

The eutrophication processes occurred most intensively in Lake Chukhlomskoe. One of the main reasons for this was the lack of commercial fishing. Novak identifies that "often there is a massive death of fish as a result of insufficient feed, due, in turn, to the high population density of individual species of fish fauna. Organic decomposition products that accumulate in water have toxic properties, cause poisoning and death of aquatic organisms. The gradual transformation of organic substances and their mineralization determine the effect of fertility, and the enhanced development of macro- and microphytes. Macrophytes and phytoplankton intensively consume oxygen, resulting in fish killing, due to a lack of oxygen dissolved in water" [11].

In addition, frequent fluctuations in the water level, reaching 1 m, had an adverse effect on inhabitants of the littoral zone and spawning fish in these areas, which could lead to the disappearance of some of the most sensitive species.

The problem of eutrophication has affected various freshwater systems in Europe and Asia [14,15].

Changes in the hydrobiocenoses of water bodies under eutrophication are manifested in several directions. The number of green algae and cyanobacteria in phytoplankton increases. Species diversity is generally declining. An important sign of eutrophication is the decrease in the number of large forms of zooplankton (Diaptomus) and their replacement by cyclops, rotifers and cladocerans [16].

At the same time, increased eutrophication of the reservoir has led to an increase in the species diversity of zoo benthos. So, in the first years of the creation of the Rybinsk reservoir on the Volga, benthos of sandy soils was represented by single individuals of chironomids, oligochaetes and leeches. In 1986 and 1990, 115 and 133 species, respectively, were recorded as a part of macro zoo benthos of open shallow water, during periods of maximum eutrophication of the reservoir [17].

Under the influence of eutrophication, the ratio of zoo- and phytoplankton biomass decreased and the value of zooplankton in total production decreased. The trophic links within the planktonic community were weakened. The degree of eutrophication of water bodies varied depending on climatic factors, such as phases of water content and temperature [18].

The littoral zones of the reservoirs were most susceptible to anthropogenic impact, due to the contact of two natural complexes: land and water. The most informative structural parameters, adequately reflecting changes in overgrown biotopes subject to anthropogenic pollution, were the total abundance and biomass of zooplankton, the abundance and biomass of copepods, branched crustaceans, the species diversity index of zooplankton biomass and the number of dominants allocated by function. In anthropogenically polluted macrophyte tangles, there was a tendency towards an increase in the quantitative parameters of crustacean zooplankton groups (*Cladocera* and *Copepoda*) and the total abundance

and biomass of these communities, as well as an increase in the degree of dominance of individual species, compared with clean areas [19].

In lakes of Karelia, dynamics of long-term changes in the composition of biocenoses similar to the Volga basin were observed. In the first period (from 1954 to 1956), the anthropogenic impact on the lake was insignificant and the process of natural eutrophication prevailed. Vendace, pike perch, perch, ruff and bream dominated in fish fauna. The second period (1973–1985) was associated with significant anthropogenic impact [20–22]. The supply of biogens sharply increased. As a result of land reclamation, the water level decreased, and the littoral zone of the lake decreased. The planktonic type of fish nutrition began to prevail over the benthic one. As a result, the number of whitefish and bream decreased. Since the 1990s, due to a decrease in the anthropogenic load, there has been a tendency to slow down the eutrophication process. Smelt, perch, ruff, pike perch and bream began to dominate in the fish fauna [14,20].

Kazakov [23] noted in the lakes of Karelia a correlation of the trophic level and the composition of the ichthyocenosis: dystrophic—perch monoculture; slightly dystrophic—perch, roach, pike; mesotrophic—perch and 6 fish species; oligotrophic—perch and 15 fish species.

The intensification of the eutrophication process is currently characteristic of many reservoirs in the European part of the Russian Federation [24–26]. Rivers with a high flow rate, a low content of organic substances and the absence of significant anthropogenic impact are classified as pure with a high variety of rheophilic oligo saprobic forms of zoo benthos. The species diversity index is 3.77–4.33. Constant anthropogenic pressure leads to river pollution, which leads to succession with the dominance of limnophilic mesosaprobic forms of macro zoo benthos, and the species diversity index decreases to 3.3 [27].

The leading factor in increasing the level of primary production is the regulation of river systems, the discharge of insufficiently treated effluents, the flushing of fertilizers and pesticides by rain and meltwater from agricultural land, and the filtrate from waste dumps [28]. Biocenoses under conditions of increasing concentration of nutrients are transformed in a similar way in both marine and freshwater ecosystems [29,30], which makes it possible to distinguish indicator species of aquatic organisms, the study of which has both theoretical and applied value.

To develop a sustainable water management system, it is necessary to organize a global system for monitoring and forecasting changes in climatic conditions, their impact on water balance, ecological systems, air quality, natural and anthropogenic biocenoses [31–33].

## 4. Conclusions

A significant accumulation of nutrients in water bodies and an increase in the level of eutrophication were noted during the regulation of the river flow and the flow of wastewater into river systems. At the same time, their trophic status changed and water quality decreased, leading to the depletion of species diversity in biocenoses.

The study of hydrological and ecological–biological parameters of reservoirs in the northern part of the upper Volga region formed the basis for classifying, according to the level of eutrophication, the structure of the populations of dominant fish species, and the species composition of the ichthyofauna. The channel part of the Gorky reservoir is a mesotrophic–eutrophic reservoir, the Kostroma spill is eutrophic–hypertrophic, Lake Galichskoe is eutrophic-dystrophic, and Lake Chukhlomskoe is hypertrophic–dystrophic.

The main causes of eutrophication were established, including: the flow violation as a result of dams construction; expansion of the surface of the water mirror; the spilling of the coastal part and leaching of mineral substances, causing a decrease in depth due to the sediment of mineral particles at the bottom of reservoirs at low-flow velocity, or the complete absence of it; the more intensive warming of water in shallow areas and near hydroelectric power plants and, as a result, the acceleration of the decomposition of organic compounds in dissolved and undissolved forms.

A high concentration of nutrients in the water has led to the hypoxia of hydrobionts. Their death, due to a lack of oxygen dissolved in the water, causes a transition from

eutrophy or hypertrophy to dystrophy, and a decrease in species diversity. These processes are most pronounced in lakes Galichskoe and Chukhlomskoe, in comparison with other studied reservoirs of the northern part of the upper Volga region.

**Author Contributions:** Conceptualization, A.N. and M.N.; methodology, B.G. and A.N.; software, D.V.; validation, D.V., B.G. and R.P.; formal analysis, R.P.; investigation, B.G., A.N. and M.N.; resources, A.N., M.N. and D.V.; data curation, B.G. and R.P.; writing—original draft preparation, B.G., A.N., M.N. and D.V.; writing—review and editing, A.N., B.G. and R.P.; visualization, A.N.; supervision, B.G., M.N. and R.P.; project administration, B.G., M.N. and R.P.; funding acquisition, B.G. and R.P. All authors have read and agreed to the published version of the manuscript.

**Funding:** This research was funded by grant of Wrocław University of Environmental and Life Sciences, Institute of Soil Science, Plant Nutrition and Environmental Protection (B010/0035/21), from the subsidy of the Ministry of Education and Science of Poland.

**Institutional Review Board Statement:** Not applicable.

**Data Availability Statement:** The study did not report any data.

**Conflicts of Interest:** The authors declare no conflict of interest.

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
