# Peer review of "Ecological Parameters of Water Bodies in the Northern Part of the Upper Volga Region with River Flow Regulations"

_water, doi:10.3390/w13243586_

Round 1

Reviewer 1 Report

Authors investigated ecological study of water bodies located in the northern part of the Upper Volga region. This study was conducted in the river flow regulation. Overall, the paper has no potential of readers' interest because authors did not follow a standard format of a journal paper for introduction. The paper after taking into account comments below:

1-Abstract section needs some main findings after presentation of Carlson Trophic Index (TSI) results for various eutrophic statuses. 

2-Introduction section has no literature review. Authors are recommended to read the most updated references from various reliable references.

A Novel Multiple-Kernel Support Vector Regression Algorithm for Estimation of Water Quality Parameters

Reliability assessment of water quality index based on guidelines of national sanitation foundation in natural streams: integration of remote sensing and data-driven models

Prediction of the five-day biochemical oxygen demand and chemical oxygen demand in natural streams using machine learning methods

3-Introduction section has no innovation, motivation, and research organization. 

4-Case study needs fundamental improvements: (i) Why did authors consider this location? (ii) Details of case study need to be discussed.

5-Conclusion section needs to be merged into revision.

6-Carlson's trophic index was calculated by the formula [Eq.(1)]. This formulation needs to be detailed for various ecological states. Additionally, number of equation is missing and the relationship needs to be re-written.

7-Figure 1 has poor quality.

Author Response

Please look at the attachment

Reviewer 2 Report

Thank you for providing your manuscript to the Water journal. Generally, the topic fits into the scope of the journal, however the content is poor. This situation results from several reasons, mainly because the literature review is extremely poor until unexisting, and the methdology is not new at all. Moreover, the figures are of poor quality, the sources are missing and the literature citations are not in line with the journal requirements (must be figures in brackets). From my point of view, the manuscript should be rejected for a couple of reasons, and the only reason why I recommed a major revision is that scientific information from that geographical part of the world is rarerly found in the english speaking scientific literature. Moreover, in the current version, the manuscript doesnt qualify for a classification as article, it might be considered a case report.

Also the methodology section must be revised throughout. There is missing the information on the way of sampling, the number of samples, their (potential) preparation and investigation strategy, etc. I strongly recommend to include a flow chart illustrating the steps of the methodology. The site description doesnt belong to the results section and should be added to the methodology section.

In the literature review, it is important that the scientific novelty of the work is established through a critical analysis of related literature. Furthermore, in the introduction must be given a general overview on the subject under review, and the motivation why the study is performed as well as the detailed scope of the study. Thus, the main questions of the reviewer are: What is the scientific motivation for the study? Which scientific question(s) shall be answered with this? What is your scientific hypothesis that you wish to answer with the investigation? Putting the scientific motivation will also help you to identify the novelties that characterises a scientific publication. Moreover, the red line in the manuscript is missing. In the current form it is just a collection of data on the respective lakes. 

In the results section, it remains completely unclear if the informtion given in the figures 2, 3, 5, 6 and 7 ist from own investigations or from the literature. The majority of the cited literature is very old.

In the conclusions, in addition to summarising the actions taken and results, please strengthen the explanation of their significance. It is recommended to use quantitative reasoning comparing with appropriate benchmarks, especially those stemming from previous work.

Author Response

Please look at the attachment

Round 2

Reviewer 1 Report

After acceptance of paper, academic editors ask authors to consider following comments:

1-In lines 59-103, introduction requires to combine.

2-In lines 113-175, "materials and methods" section requires combinations

Author Response

The authors' response

We would like to thank the esteemed reviewer once again for re-analyzing the revised manuscript of the article.

According to the current comments, we can answer that the paragraphs in the sections "Introduction" and "Materials and Methods" are combined in meaning, redundant references to the source of citation are removed.

Reviewer 2 Report

Thank you for providing the revised version. Some improvement has been made, however, there are still a lot of issues with the manuscript.

Even the state of the art was improved (literature research) this part is still weak. There is stil recent literature that has not beed cited yet and must be added, for example:

https://onlinelibrary.wiley.com/doi/abs/10.1002/rrr.3450110107

In this regard I still strongly recommend to improve the literature research.

Moreover, the overview map for the location of the investigation area should be moved to the materials and methods section. The quality of the maps must be improved. A flow chart should be added that illustates the steps of investigation.

The information on Trophic index and related parameters is very old (1977). recent data must be added.

From my point of view, scientific proof must be added that the autrophication results from dam construction.

Author Response

The authors' response

Dear reviewer, we would like to thank you for the repeated detailed analysis of our manuscript. Your opinion is very valuable to us as a highly qualified scientist, a specialist in the field of hydrobiology. We have tried to take your recommendations into account as much as possible.

Thank you for the link to the research of scientists around the world on the regulation of river flow, changes in ecosystems after the construction of dams. We have given in the "Introduction" section several works that vividly illustrate the problem described in our article.

It was decided to completely remove the cartographic material from the article. The section "Materials and methods" describes the research area in sufficient detail. In the same section, a diagram of the completed studies has been added in the form of a table.

I would like to discuss Carlson's trophic index. This indicator is still estimated according to the formula proposed by Carlson in 1977. Despite the fact that it was proposed about 50 years ago, it has not lost its relevance. The water transparency indicator vividly illustrates the processes of eutrophication. Therefore, we left these calculations and supplemented the results with ITS calculations for Broth (1982), based on the concentration of chlorophyll in the upper layer of water.

According to your last remark: "From my point of view, scientific proof must be added that the autrophication results from dam construction", we would like to note that scientifically based evidence of the influence of dams on eutrophication processes is given by us in the article and confirmed by the research of other authors, which we refer to in the sections "Introduction" and "Results and discussion".